# Breath and Sputum Analyses in Asthmatic Patients: An Overview

**DOI:** 10.3390/cells13161355

**Published:** 2024-08-14

**Authors:** Piera Soccio, Carla Maria Irene Quarato, Pasquale Tondo, Donato Lacedonia, Anela Hoxhallari, Maria Pia Foschino Barbaro, Giulia Scioscia

**Affiliations:** 1Department of Medical and Surgical Sciences, University of Foggia, 71122 Foggia, Italy; pasquale.tondo@unifg.it (P.T.); donato.lacedonia@unifg.it (D.L.); dr.anelahoxhallari@gmail.com (A.H.); mariapia.foschino@unifg.it (M.P.F.B.); giulia.scioscia@unifg.it (G.S.); 2Institute of Respiratory Diseases, Policlinico Riuniti of Foggia, 71122 Foggia, Italy; carlamariairene.quarato@gmail.com

**Keywords:** sputum, EBC, FeNO, non-invasive methods, airway inflammation, asthma management

## Abstract

Recent advancements in asthma management include non-invasive methodologies such as sputum analysis, exhaled breath condensate (EBC), and fractional exhaled nitric oxide (FeNO). These techniques offer a means to assess airway inflammation, a critical feature of asthma, without invasive procedures. Sputum analysis provides detailed insights into airway inflammation patterns and cellular composition, guiding personalized treatment strategies. EBC collection, reflecting bronchoalveolar lining fluid composition, provides a non-invasive window into airway physiology. FeNO emerges as a pivotal biomarker, offering insights into eosinophilic airway inflammation and aiding in asthma diagnosis, treatment monitoring, and the prediction of exacerbation risks. Despite inherent limitations, each method offers valuable tools for a more comprehensive assessment of asthma. Combining these techniques with traditional methods like spirometry may lead to more personalized treatment plans and improved patient outcomes. Future research is crucial to refine protocols, validate biomarkers, and establish comprehensive guidelines in order to enhance asthma management with tailored therapeutic strategies and improved patient outcomes.

## 1. Introduction

Asthma is a chronic disease of the airways characterized by bronchial obstruction, usually reversible spontaneously or following therapy, bronchial hyperresponsiveness, and accelerated decline in respiratory function that can evolve into irreversible airway obstruction in some cases [1].

Numerous mechanisms are involved in the pathogenesis of these alterations, particularly the infiltration of inflammatory cells, release of mediators, and airway remodeling [2]. Clinically, it manifests with dyspnea, wheezing, cough, and a sense of chest tightness, whose intensity varies in relation to the extent of bronchial obstruction and the degree of its perception by the patient [1].

The severity of asthma’s clinical manifestations is generally correlated with the extent of bronchial obstruction but can be perceived differently by different individuals or in different phases of the disease, making asthma an extremely heterogeneous condition [3].

The physiological and clinical characteristics of asthma result from an interaction among inflammatory cells, inflammation mediators, and cytokines at the epithelial surface level. Cells that play an important role in the inflammatory response include mast cells, eosinophils, lymphocytes, and epithelial cells [4]. Each of these cell types can produce mediators and cytokines that trigger and amplify both acute inflammation and long-term alterations. Mediator release causes an intense and immediate inflammatory reaction leading to bronchoconstriction, vascular congestion, edema formation, increased mucus production, and impairment of mucociliary clearance [4].

As established by the Global Initiative for Asthma (GINA) Guidelines, both asthma diagnosis and disease monitoring are based on symptom collection, physical examination, and respiratory function tests (spirometry, reversibility testing, and nonspecific provocation testing), as well as investigations to identify risk factors such as atopy and the familial history of allergic or pulmonary diseases [1]. Although asthma is a disease supported by a chronic inflammatory process of the airways, its role in the formulation of diagnosis and, more generally, in the management of the disease for indirectly determined methodologies of the bronchial inflammation level, such as sputum analysis, exhaled breath condensate analysis, and/or nitric oxide measurement in exhaled air, is still not fully recognized. However, over the last decade, substantial scientific evidence has been obtained regarding the possibility of employing non-invasive methods for measuring airway inflammation in clinical practice [5,6,7,8]. Among these, the search for eosinophils in spontaneous or induced sputum is one of the simplest, safest, and most reliable tests for monitoring airway inflammation [9]. The concentration of eosinophils in sputum can be used as a parameter in the management of asthmatic patients and to assess whether an improvement in symptoms and an increase in therapy correspond to a reduction in eosinophil concentrations in sputum. Breath analysis allows a rapid non-invasive diagnosis of respiratory diseases by identifying and quantifying exhaled biomarkers. The study of the exhaled nitric oxide fraction (FeNO) enables the determination of airway inflammation [10]. Nitric oxide (NO) is indeed an inflammation marker, and its concentration in exhaled air can be used to assess the degree of airway inflammation. In particular, in asthma, FeNO has been studied as a response marker for inhaled corticosteroid (ICS) therapy and as a marker for the effect of pollutants on airway inflammation. Likewise, exhaled breath condensate (EBC), a biological sample derived from the condensation of epithelial lining fluid, represents a useful matrix for dosing and monitoring inflammation mediators such as prostaglandins, leukotrienes, cytokines, and growth factors [11], as well as for the dosage and study of oxidative stress mediators and pH analysis. These measurements reflect significant components of the pathophysiology of this disease, such as eosinophilic inflammation and host defenses during exacerbations [12]. Based on these considerations, this review aims to evaluate the benefits of adding to the clinical and functional evaluation parameters of asthma established by international guidelines, the search for eosinophils in sputum, the analysis of exhaled breath condensate, and the study of the exhaled nitric oxide fraction (FeNO) as indicators of the degree of bronchial inflammation.

## 2. Sputum

Sputum is a substance secreted by the respiratory tract, particularly from the bronchi and lungs. It mainly consists of water, mucins, epithelial cells, cellular debris, bacteria, inflammatory cells, and other components. Sputum production is a natural response of the body to protect the respiratory tract from irritation, infection, and other aggressions. Sputum collection occurs through both spontaneous and induced production, although it has been demonstrated that most spontaneously obtained samples were of poor quality and that not all patients were able to produce sputum [13,14,15]. On the other hand, induced sputum represents a more reliable method compared to spontaneous collection because it allows the controlled production of higher-quality samples, enabling the assessment of airway inflammation and the investigation of new therapies in a non-invasive and repeatable manner in the context of lung diseases, making it suitable for large-scale studies and clinical trials with multiple visits [16,17]. Since 1999, the methodology for collecting and processing sputum has been standardized by the guidelines of the European Respiratory Society (ERS), allowing for the global comparison of data and improving the quality and reproducibility of sputum samples [18]. In asthma management, the GINA guidelines recommend analyzing spontaneous or induced sputum to assess airway inflammation. Measuring the levels of inflammatory cells in sputum, such as eosinophils or neutrophils, allows for more precise and individualized treatment adjustment, thereby improving asthma symptom control and reducing the risk of exacerbations [1,19]. Furthermore, as reported by the GINA guidelines, since induced sputum measures the degree of bronchial inflammation, it can distinguish between different asthma phenotypes [1,20]. Standardizing the sputum collection methodology, along with the GINA guidelines supporting its use in asthma, helps improve disease management and optimize clinical outcomes for patients with this chronic respiratory condition.

### 2.1. Sputum Induction and Processing Procedure

Sputum induction is a crucial procedure in the treatment of patients unable to spontaneously produce sputum, especially those with conditions such as asthma. This process involves the inhalation of a saline solution, with sodium chloride concentrations ranging from 0.9% to 7%, followed by the expectoration of airway secretions through coughing. Although various concentrations of saline solution (ranging from 0.9% to 7% [21,22,23]) have been tested, the standard concentration recommended by the European Respiratory Society Task Force is 4.5% [18]. However, sputum induction has the potential to induce airway inflammation [24] and bronchoconstriction [17] in asthmatics, which can lead to even fatal asthma exacerbations in subjects with severe airflow obstruction [25]. For patients at risk of bronchoconstriction (baseline FEV1 after bronchodilator < 60% of the predicted level), isotonic saline solution (0.9%) is recommended as an alternative to the hypertonic saline solution [18]. Hypertonic solutions have been shown to be more effective in inducing sputum; however, no significant difference in the cellular composition between isotonic and hypertonic solutions has been found [21,22,26]. During induction, the use of ultrasonic nebulizers is recommended, which have demonstrated greater efficacy in producing mucus samples compared to jet nebulizers [27] (Figure 1).

One critical aspect during sputum induction is the duration of inhalation. Throughout the sputum induction procedure, various compartments of the respiratory tract are sampled: the central airway, followed by the peripheral airway, and finally, the alveoli. Maintaining a consistent duration of inhalation ensures more accurate comparisons between subjects. It has been shown that shorter (15–20 min) and longer (30 min) inhalation durations exhibit the same success rate and feasibility in sputum production [26]. Furthermore, research has shown that the average percentage of various types of cells remains largely unchanged in samples collected consecutively after 5, 10, and 15 min, suggesting that a 15-min induction procedure, using a consistent concentration of the hypertonic saline solution, interspersed with coughing and expectoration every 5 min, along with lung function measurement to detect bronchoconstriction and the analysis of the mixed sample, could be advisable [18,28,29] (Figure 2).

Before proceeding with sputum induction, it is crucial to perform a respiratory function check on the patient using spirometry. This preliminary step is necessary to identify potential risks of airway constriction in individuals with asthma when exposed to saline solutions during treatment. To prevent such complications, all patients receive a preventive treatment with a rapid-action bronchodilator, typically, beta-2 agonists, before the start of induction. A preventive dose of salbutamol, generally between 200 and 400 µg, is administered through a pressurized inhaler as a measure of bronchial protection without altering the analysis of cellular components or inflammation markers [17,30]. This procedure must be carried out under the supervision of qualified medical personnel and in the presence of a physician, who can promptly intervene in case of adverse reactions. The sample processing phase of sputum represents a crucial step in the analysis of collected samples. Currently, there are two main methods for sputum processing that present slight variations, both of which are accepted by international ERS guidelines [18]. One method involves selecting “plugs,” i.e., dense portions of sputum, using forceps and an inverted optical microscope to minimize saliva contamination. This approach can be used to obtain better-quality material for sputum cytological studies and a more reliable concentration of biomarkers in the supernatant. However, this procedure requires more time and the use of an inverted microscope [5,9,31]. The other suggested approach involves analyzing the entire sputum sample, including saliva. This method undoubtedly offers the advantage of greater speed compared to the former; however, the presence of saliva can lead to sample dilution and compromise the accuracy of the analysis. Consequently, samples processed in this way often exhibit a high quantity of squamous cells, making the counting of inflammatory cells complex, especially if squamous cells constitute more than 20% of the total cells [5,9,15,32,33]. The sample must be processed within 2 h of collection to ensure accuracy in cell counting and staining. After selection, the sample is treated with dithiothreitol (DTT), a mucolytic agent that breaks the disulfide bonds present in mucus and allows the release of cells trapped in the mucosal matrix [34,35,36]. Subsequently, the solution is filtered through a nylon mesh filter (53 µm mesh) to remove mucus and debris, thus optimizing the quality of smears without influencing the differential cell count, [13,34,35,36,37]. The filtered sample is then centrifuged at 600× *g* for 10 min at 4 °C–8 °C [18] and the supernatant is aspirated and stored at −80 °C for further analysis, while the corresponding cell pellet is resuspended in phosphate buffer and used to perform the total cell count using a hemocytometer (Figure 3).

Additionally, the pellet is used to assess cell viability to obtain information about the activity of the inflammatory process in the patient’s airways. Sample viability is evaluated using Trypan blue dye, which penetrates only damaged cell membranes. The differential cell count is performed by preparing cytospins stained with May-Grunwald Giemsa or Diff-Quick stain [34,35,36]. The differential cell count under the optical microscope is performed by counting at least 400 non-squamous cells. Squamous cells, instead, are counted separately as an indicator of saliva contamination in the sample [34,35,36].

To properly understand sputum cell results, it is essential to be familiar with the normal range of the cell count in induced sputum from healthy nonsmoking adults. Studies have shown that most cells are macrophages and neutrophils, while eosinophils, lymphocytes, and bronchial epithelial cells are rare, and metachromatic cells (basophils/mast cells) are almost entirely absent [38]. Deviations from these normal values can provide valuable insights for healthcare professionals, guiding accurate diagnoses and personalized treatment plans (Table 1).

### 2.2. Sputum Analysis: Unveiling Insights into Asthma

Sputum, or phlegm, is a readily accessible and non-invasive biological matrix that provides a broad insight into respiratory conditions. Its non-invasive nature, low cost, and widespread acceptance by patients make it an ideal and versatile sample for various clinical and research applications. In clinical practice, induced sputum is used to assess airway inflammation in patients with conditions such as asthma, chronic cough, and chronic obstructive pulmonary disease (COPD) [39]. The analysis of sputum cell composition, particularly the differential cell count, provides crucial information on airway inflammation and can be helpful for diagnosis and monitoring treatment response [40]. The ability to identify and quantify the different cellular populations present in sputum allows physicians to assess asthma-related specific types of airway inflammation. An elevated percentage of eosinophils in induced sputum (i.e., >2%) is included in the current GINA guidelines among the criteria for identifying type-2 asthma [1]. This correlation is essential for the diagnosis and monitoring of asthma as it provides insights into the inflammatory nature of the condition and the effectiveness of anti-inflammatory drugs in reducing airway inflammation [13,41]. In particular, elevated levels of eosinophils in sputum have been associated with the severity of the disease and the response to treatment with inhaled corticosteroids in type-2 asthmatic patients [40]. In clinical trials evaluating the effectiveness of monoclonal agents, no reduction in sputum eosinophils was found during treatment with omalizumab [42] and mepolizumab [43] in severe type-2 asthmatics. However, Mukherjee et al. [44] found suboptimal responses to mepolizumab in cases of persisting sputum eosinophilia despite blood eosinophil count normalization. In contrast, in the real-life PROMISE study [45], a significant reduction in blood and sputum eosinophil counts was associated with an important and sustained reduction in the exacerbation rate and oral corticosteroid (OCS) dose in patients with severe eosinophilic asthma receiving benralizumab. In the recent clinical trial by Svenningsen et al. [46], dupilumab reduced sputum eosinophils and CT mucus plugging, allowing a consequent improvement in MRI ventilation and asthma control. Furthermore, tezepelumab addition reduced levels of blood and sputum eosinophils and allergen-induced bronchoconstriction in a population of patients with mild allergic asthma before and after allergen challenge [47]. Overall, the use of induced sputum as a tool for evaluating airway inflammation represents a significant advancement in clinical practice, enabling more accurate diagnosis and more effective management of asthma.

Additionally, the analysis of induced sputum supernatant has proven to be a powerful research tool, shedding light on a wide range of respiratory conditions. In the context of asthma research, it allows the exploration of numerous biological markers of airway inflammation and specific immune responses, providing valuable insights into the pathogenesis, severity, and treatment response of asthma [41,48]. Among the main biomarkers studied in induced sputum for asthma are cytokines and inflammatory mediators. The presence and concentration of proteins such as eosinophilic granule protein-2 (EG-2), granulocyte-macrophage colony-stimulating factor (GM-CSF), tumor necrosis factor-alpha (TNF-α), and interleukin (IL)-8 have been correlated with airway inflammation and asthma severity [15,49,50]. These measurements are generally performed using the immunoenzymatic method. However, DTT may interact with commercial ELISA kits or alter the three-dimensional structure of proteins dissolved in mucus, thus limiting the measurement of soluble mediators in induced sputum. Previous studies of induced sputum from asthmatics have shown that processing with DTT may decrease detectable concentrations of myeloperoxidase (MPO) and eosinophilic peroxidase and increase concentrations of eosinophil cationic protein (ECP), but it does not seem to have any effect on IL-5 or IL-8 levels [34,35,51]. Over the past decade, the combination of electrophoretic/chromatographic methods and mass spectrometry has gained growing interest in the identification of proteins whose levels change in the sputum of patients with asthma. Using the mass spectrometry technique, Lee et al. [52] found that the expression of S100 calcium-binding protein A9 (S100A9) was significantly increased in the sputum of severe uncontrolled asthmatic patients compared to controlled ones and was associated with neutrophilic inflammation. According to Takahashi et al. [53], different expressed proteins in induced sputum could make it possible to distinguish between a smoking and ex-smoking severe asthma phenotype. Tariq et al. [54] found that proteins associated with gastro-oesophageal reflux disease were three- and ten-fold more prevalent in the sputum of severe asthmatics compared to that of mild/moderate asthmatics and healthy controls. In addition to cellular markers and cytokines, research has also focused on gene expression and microRNA (miRNA) profiling in induced sputum from asthmatic individuals. These small RNAs have been identified as key regulators of gene expression and may be implicated in asthma pathogenesis. Several studies have revealed specific miRNA expression patterns associated with the severity and phenotype of asthma, thus providing valuable insights into the underlying biology of the disease and opening new avenues for the development of targeted and personalized therapies [55,56,57,58,59,60,61,62]. Gomez et al. [55] identified hsa-miR-223-3p as the highest expressed microRNA in the sputum of asthmatics with increased neutrophil counts, thus suggesting that it is a key regulator of the TLR and Th17 pathways. Li et al. [57] linked an up-regulation of miR-9 to the mechanism of steroid resistance in asthma. According to Zhang et al. [58], a decreased expression of miR-221-3p in sputum may protect asthma patients against eosinophilic airway inflammation. Lacedonia et al. [59] found that miRNA-145, a key regulator of airway smooth muscle (ASM) function, is higher in the sputum of asthma patients compared to those of patients with different obstructive diseases. Malmhäll et al. [60] suggested a possible role of miR-155 expression in allergen-induced eosinophilic inflammation as its expression was decreased in the sputum of allergic asthmatics during the pollen season. According to the suggestive hypothesis of Song et al. [61], an up-regulation of lncRNA-NEAT1 and a down-regulation of miR-128 in the sputum of children with bronchial asthma may promote airway smooth muscle remodeling. Briefly, understanding the patterns of gene expression and miRNA in sputum could lead to the development of new therapeutic approaches for asthma, aimed at modifying the specific pathogenetic pathways identified through these molecular analyses. The analysis of sputum cellular composition, along with the assessment of molecular markers, provides an in-depth understanding of asthma pathogenesis and severity, offering fundamental data to personalize therapies and develop new targeted treatments, thereby improving clinical management and the quality of life of asthmatic patients (Figure 4).

## 3. Exhaled Breath Condensate (EBC)

Exhaled breath condensate (EBC) is a non-invasively obtainable biological matrix resulting from the condensation of epithelial lining fluid, useful for studying pathological processes in the airways [8]. The exhaled air leaving the mouth has a temperature of about 35 °C and a humidity of 95%. Exhaled air is almost entirely in equilibrium with water vapor at body temperature. When the exhaled air impacts a surface cooler than the temperature of water vapor, condensation occurs. Therefore, EBC is a liquid matrix essentially composed of condensed water vapor [63] (Figure 5).

Numerous volatile and non-volatile biologically active substances can be determined in EBC [64]. The mechanism by which exhaled substances are found in the condensed breath is not entirely clear. It has been hypothesized that small particles, probably droplets detaching from the liquid film lining the respiratory tract, remain suspended in the exhaled air and are transported outward by the exhaled vapor stream.

To date, the measurement of inflammatory markers in exhaled breath condensate has added a new perspective to the study of airway inflammation [65]. Currently, the most studied markers in EBC are interleukins (IL-1β, IL-8, IL-10, and IL-6), tumor necrosis factor-alpha (TNF-α), and pH. EBC pH is simple to measure, non-invasive, economical, and reproducible, making it an excellent marker of airway inflammation. Markedly acidic pH levels have already been measured in exhaled breath condensate of adults with asthma, chronic obstructive pulmonary disease (COPD), and bronchiectasis [12].

### 3.1. Collection and Analysis of EBC

Collecting exhaled breath condensate is a particularly attention-demanding method as it does not alter the airway mucosa and does not alter the structure and functional status of the lower airways. The airway lining fluid component of EBC is diluted by condensing vapor-phase water in an unpredictable ratio. As a result, it is assumed that to calculate the “real airway level” of determined mediators in EBC, it is crucial to determine the so-called “dilution factor” of each sample. To this end, some authors have suggested using concentrations of electrolytes [66,67,68] or urea [69] or the measurement of conductance [67], but these methods have not yet been used extensively and standardized. Nonetheless, the dilution of EBC samples is less variable than those obtained by bronchoalveolar lavage (BAL) as no external fluid is added to the airways [70]. To date, published data on inflammation mediators are consistent with the anomalies observed using bronchoscopy, suggesting that EBC could be a reliable method to assess the inflammatory status of the airways [71]. Based on the above, it is essential to emphasize that EBC collection provides information on the pathophysiological mechanisms in the airways by detecting changes in mediator levels and provides information on the composition of bronchoalveolar lining fluid. The collection is performed by asking the subject to breathe through the mouth at tidal volume for a predetermined time using a device (condenser) that cools the exhaled air, allowing the collection of the condensate, a biofluid whose composition mirrors that of the surface liquid in the airways [72]. The volume of EBC obtained in 15 min of tidal volume is approximately 2 mL [73]. The most used techniques for the determination of different biomarkers are colorimetric and immunoenzymatic. However, to achieve greater analysis specificity, methods combining chromatography and mass spectrometry are often used [74]. For most substances determinable in EBC, determination can be performed on the unmanipulated sample. However, this is not always possible, and for the analysis of some analytes, sample processing is required before analysis. For pH, for example, it is necessary to introduce an inert gas into the EBC sample for several minutes to remove volatile substances (e.g., CO_2_) and determine fixed acidity [12]. In healthy conditions, a standardized EBC sample (i.e., de-aerated with a CO_2_-free gas such as argon, nitrogen, or oxygen) has a mean pH of 7.7, with a normal pH range of 7.4–8.8 [75]. For other substances, such as some cytokines, the determination of the dilution factor is often necessary.

### 3.2. Exhaled Breath Condensate (EBC) in Asthma: Challenges and Future Directions

EBC collection is a method for sampling airway lining fluid that can be performed multiple times without harming the patient. It represents a simple and entirely non-invasive procedure that can contribute to a better understanding of asthma pathophysiology. Numerous substances can be evaluated in EBC [76], including molecules of modest dimensions, such as hydrogen peroxide, and molecules of larger dimensions, such as leukotrienes, prostaglandins, cytokines, isoprostanes, tumor markers, and small amounts of DNA. Despite the encouraging positive results obtained so far, the introduction of EBC into routine clinical practice requires resolving some methodological pitfalls, standardizing EBC collection, and, ultimately, identifying a reliable biomarker that is reproducible, has normal values, and provides information regarding the underlying inflammatory process and treatment response. So far, none of the parameters studied in EBC meets the requirements, with one possible exception: pH. This parameter is reproducible, has normal values, reflects a significant portion of asthma pathophysiology, and is measurable on site with a standardized methodology [75]. The pH in EBC is reduced in cases of persistent asthma, lowered in exacerbations, and correlated with disease severity and smoking [77,78,79]. An up to three log order decrease in the EBC pH of asthmatic patients reflects eosinophilic inflammation and nitric oxide (NO) metabolism [80,81]. Furthermore, the development of exercise-induced bronchoconstriction was related to an acute reduction in EBC pH [82]. Similarly, elevated concentrations of EBC Cys-LTs and LTB4 were found in asthmatic children during exacerbations, significantly decreasing after prednisone treatment [83], and EBC Cys-LT values have been found to increase in asthmatic children with exercise-induced bronchoconstriction [84]. Another promising marker of airway inflammation in EBC seems to be the adenosine concentration, whose value has been shown to be higher in patients with worsening asthma symptoms or with exercise-induced bronchoconstriction [85,86]. The EBC adenosine concentration showed a positive correlation with FeNO levels in asthmatic patients [85]. Based on the above, EBC analysis and levels of exhaled fractional nitric oxide (FeNO) are the only methods that allow the in vivo investigation of airway inflammation associated with the development of bronchoconstriction. However, due to the lack of standardization of the procedure, the pH, Cys-LTs levels, and adenosine concentration in EBC have not been prospectively evaluated as a guide for asthma treatment, such as exhaled nitric oxide and/or eosinophils in sputum. Based on these considerations, in addition to evaluating new biomarkers, it is necessary to standardize existing procedures for the introduction of EBC into clinical practice.

To date, few recommendations have been published to facilitate standardization in the EBC collection and analysis methodology. In 2005, the American Thoracic Society (ATS) and the European Respiratory Society (ERS) created a task force to determine guidelines for collecting and analyzing exhaled breath condensate [87]. However, it was only in 2017 that recommendations for the EBC collection procedure and technical analysis standards were published by the ATS/ERS as a guide for studies in this regard [88]. Since then, numerous airway inflammation mediators and oxidative and nitrosative stress markers such as pH, hydrogen peroxide, isoprostanes, cytokines, leukotrienes, prostanoids, nitric oxides, and peptides have been studied in EBC samples from patients with lung diseases [89,90,91,92]. In this context, our study group examined periostin concentrations in the airways of severe asthma patients, particularly focusing on the type-2 (T2) immunity endotype, using EBC and induced sputum [93]. The findings revealed elevated periostin levels in severe asthma patients compared to those with mild to moderate asthma and healthy controls. Specifically, periostin levels were significantly higher in the T2 endotype compared to non-T2 endotypes. This highlights periostin in the airways as a valuable marker for identifying severe eosinophilic asthma patients likely to benefit from biologic therapies [94]. Recently published data have also shown that EBC could be a minimally invasive, economical, and rather efficient method for detecting biological macromolecules such as microRNAs (miRNAs) [94]. These small molecules regulate gene expression and influence various biological processes underlying asthma pathogenesis, such as the inflammatory process, remodeling, and exacerbation of airway obstruction [95,96,97,98,99]. The use of miRNA as diagnostic markers in assessing asthma exacerbations is increasingly gaining ground in the management of asthmatic patients [100]. Thanks to cutting-edge approaches, these small non-coding RNA molecules can now be detected in EBC, a non-invasive and easily obtainable biological sample, as well [100]. Interestingly, the pattern of microRNA expression in the EBC of patients with asthma seems to suggest a dysregulation of the Th2 pathway, with excessive activation of Th2 cells and secretion of Th2-type cytokines, including IL-4, IL-5, and IL-13, compared to patients with COPD and healthy adults [96]. The negative association found between symptomatic asthma and levels of the regulators miR-21-5p and miR-155 in the EBC of asthmatic children strengthens this hypothesis [97].

However, EBC still has some limitations, which are mainly attributable to the different tests used for determination, the different devices used in the collection, and the wide intra- and inter-test variability. Therefore, it is essential to conduct further studies to improve reproducibility, develop more sensitive and specific tests, and establish standard values for all laboratories. For example, EBC can be collected as a fluid at a condensation temperature of around 0 °C or as frozen material at lower temperatures. Nonetheless, it should be highlighted that the solubility of volatile mediators in the collected samples may be influenced by the collection temperature. More specifically, cold temperatures that cause the EBC to freeze may diminish the amount of volatile compounds (which are more readily absorbed into the liquid phase), while frozen storage may protect reactive or unstable compounds [75]. The collection temperature may also be clearly influenced by that of the exhaled air. The need to determine the so-called “dilution factor” to obtain the actual concentration of certain mediators in the EBC has already been discussed above. Furthermore, it must be noted that in EBC collection, there is some contamination between the saliva and the lower airways, even though studies have shown that the electrolyte ratios and the molecular content of EBC differ from those of saliva [66,101]. A frequently used method to exclude oral contamination is the detection of amylase, which, in EBCm shows levels approximately 10,000 times lower than those in saliva [66,85]. However, this measurement is not specific to salivary amylase, and amylase can also be found in the lungs. As a result, high amylase levels do not necessarily mean salivary contamination. All future publications on the use of exhaled breath condensate should contain a meticulous description of the protocol used to obtain, preserve, and analyze EBC. Only then, EBC can be considered a promising biological material to better understand the pathology and management of asthma. Having said that, it is necessary to reiterate that in a complex disease like asthma, where there is still much to discover, individual biomarkers are often not sufficient to demonstrate significant differences between asthmatic and non-asthmatic subjects [102]. Often, there is a need for more discriminant analyses that consider various biomarkers to highlight significant differences in asthmatic patients with reasonable sensitivity and specificity. Of course, this approach requires more time and greater economic resources. However, to date, this seems to be the only feasible solution. To study the degree of inflammation and bronchial obstruction in asthmatic subjects, it is necessary to simultaneously analyze different disease biomarkers. This reasoning also applies to all biomarkers detectable in EBC. The validity of EBC for the diagnosis of asthma is thus linked to the search for not a single marker but a combination of disease markers that are useful for the diagnosis and prognosis of asthmatic subjects. Therefore, until a unique and infallible biomarker is identified, the only solution is to look at a complex set of biomarkers capable of reflecting the complexity of the asthmatic condition (Figure 6).

In conclusion, despite the current challenges, the potential of exhaled breath condensate (EBC) in asthma research remains promising. The non-invasive nature of EBC collection makes it a valuable tool for studying airway inflammation and exploring novel biomarkers. However, to integrate EBC into routine clinical practice, the standardization of collection methods and validation of biomarkers is essential. Future research efforts should focus on improving reproducibility, enhancing sensitivity, and establishing universal reference values. Only through rigorous protocol adherence and comprehensive biomarker panels, EBC can realize its potential as a reliable diagnostic and prognostic tool in asthma management.

## 4. Fractional Exhaled Nitric Oxide (FeNO)

Nitric oxide (NO) is a gaseous compound with a very short half-life (i.e., equal to approximately 4 s), can penetrate through the cell membrane, and acts as an endogenous mediator in a broad spectrum of important physiological processes. In the lungs, NO is produced by different cell types, such as epithelial, endothelial, inflammatory, and nervous cells. The synthesis of this compound is entrusted to a group of enzymes belonging to the nitric oxide synthase (NOS) family, which use the amino acid L-arginine as a substrate [103]. There are two functional isoforms of this enzyme: an inducible form (iNOS) and a constitutively expressed form (cNOS). Under physiological conditions, cNOS through a Ca2+-dependent mechanism produces low levels of NO, which takes part in important physiological processes such as lung development, the relaxation of smooth muscle cells, ciliary motility, and the protection of the bronchi from bronchoconstrictor stimuli. In pathological conditions, interleukin-4 (IL-4) and interleukin-13 (IL-13), key cytokines of type-2 inflammation, through the activation of iNOS, increase the production of NO by the bronchial epithelium through a Ca2+-independent process. Part of the nitric oxide produced by iNOS, reacting with the reactive oxygen species produced by inflammatory cells (i.e., macrophages, neutrophils, and eosinophils) present in the airways, causes nitrosative stress, which results in cellular damage, altered protein functionality, and hyperreactivity of the airways [103,104] (Figure 7).

In patients with type-2 asthma, levels of exhaled fractional nitric oxide (FeNO) are elevated and correlate with other markers of disease activity, including airway hyperresponsiveness and bronchodilator response and symptoms [105]. Measuring FeNO levels is a simple, well-tolerated, and non-invasive method to evaluate airway inflammation. The current GINA recommendations [1] include FeNO among the key biomarkers for recognizing a patient with severe type-2 asthma.

### 4.1. Measurement of FeNO Levels

FeNO can be measured non-invasively using chemiluminescence analyzers or electrochemical devices. In chemiluminescence analyzers, the reaction between the NO in the sample and the ozone generated by the instrument determines the emission of electromagnetic radiation, with a wavelength between 600–3000 µm. These electromagnetic radiations are then detected and amplified, providing a signal proportional to the concentration of NO. Electrochemical sensors convert the NO concentration into an electrical signal in the presence of a buffer system and a catalytic sensor. A chemical reaction induces a quantifiable physical change in the sample that is directly proportional to the NO concentration [106]. Studies have shown that absolute FeNO measurements between these two types of devices may even differ in a clinically relevant extent [107]. To avoid discrepancies, the chemiluminescence method is currently considered the gold standard in FeNO analysis [108]. The American Thoracic Society (ATS) and the European Respiratory Society (ERS) have also developed a highly standardized procedure to measure FeNO [108]. Briefly, the subject must inhale to full lung capacity with purified air so as not to contaminate the sample with potentially high levels of NO in the environment. Subsequently, he has to exhale for 10 s at a pressure of 5–20 cmH_2_O, which guarantees the closure of the soft palate, minimizing the risk of contamination of the paranasal sinuses with NO. An exhalation is considered adequate if a stable FeNO concentration level (plateau) is reached at a given constant expiratory flow. For the given expiratory flow, three measurements are carried out and the arithmetic mean is calculated to obtain an accurate and repeatable FeNO value. The software also provides a visual encouragement, which is used to obtain the expiratory flow necessary for the measurement. An important limitation of the “standard” measurement is that FeNO appears to be strictly dependent on the amount of expiratory flow due to the diffusion of NO in the airways and the transit time: the greater the expiratory airflow (which involves a shorter duration of exhalation), the lower the FeNO, and vice versa [109,110]. Nevertheless, the concentration of NO exhaled at high expiratory flows would allow information mainly on the alveolar region, while lower flows would allow sampling of the bronchial compartment. In fact, the limited duration of exhalation at high flows (at least 200 mL/s) reduces the time during which the air remains inside the airways, limiting the diffusion of the NO produced at the level of the bronchial mucosa into the lumen, which it would thus derive to a greater extent from the alveolar compartment. At low flows, the duration of exhalation is longer and the air coming from the pulmonary alveoli remains within the airways for a sufficient time to allow the diffusion of NO from the bronchial mucosa to the lumen. For very low expiratory flows (about 10 mL/s), FeNO is derived almost exclusively from the bronchial compartment [109,111,112]. In theory, exhalation at numerous different constant expiratory flows would allow for the most precise estimation of FeNO values along the entire bronchial tree, but this is often impossible to carry out in clinical practice, both in terms of the time required for execution, and, above all, for the extent of the commitment to which patients would be subjected [112]. ATS/ERS guidelines thus recommend an expiratory flow rate of 50 mL/s (FeNO_50_) based on the hypothesis that it is more favorable to target the lower respiratory tract as the region of interest for NO excretion [108] (Figure 8).

In all chemiluminescence analyzers, the expiratory flow rate can be modified by resistors. On the contrary, electrochemical sensors are generally not suitable for the analysis of multiple flows.

### 4.2. FeNO Measurement: Confounding Factors

FeNO50 levels can also be influenced by many endogenous and exogenous factors. During childhood, FeNO_50_ physiologically increases with age and height, peaking in adulthood [113]. The male sex is associated with increased FeNO_50_ levels compared to the female sex [114]. Furthermore, in females, FeNO_50_ levels appear to vary according to the phase of the menstrual cycle, showing higher values during the proliferative or follicular phase and lower values during the secretory or luteal phase [115]. Race and ethnicity also influence FeNO_50_ levels, which are higher in healthy Hispanic and black individuals [116]. FeNO_50_ levels can decrease after exposure to tobacco smoke [117] and consumption of coffee [118], and they can increase after recent physical exercise [119] or the intake of nitrate-rich foods, such as lettuce, spinach, beets, walnuts, peanuts, and animal offal [120]. Inhaled corticosteroids (ICSs), which are the mainstay of asthma treatment, act by reducing FeNO_50_ [121]. Therefore, to eliminate confounding factors, it is advisable to perform FeNO50 measurements before other function tests (i.e., spirometry and walking tests) and to advise the patient not to refrain from eating, drinking, and smoking at least 1 h before carrying out the exam, as well as not to take their asthma maintenance therapy on the day of the exam. Elevated FeNO_50_ levels may also be found in other inflammatory conditions, such as systemic lupus erythematosus [122], liver cirrhosis [123], and COPD [124]. Furthermore, severe asthmatic patients may also be affected by other comorbidities that can influence FeNO_50_ levels. More specifically, FeNO_50_ is increased in cases of allergic rhinitis [125,126], atopic dermatitis [126], and chronic rhinosinusitis with nasal polyposis (CRSwNP) [127], while FeNO_50_ levels are low in the presence of obesity [128], obstructive sleep apnea syndrome (OSAS) [129], gastroesophageal reflux disease, and bronchiectasis [130]. Viral infections may lead to increased FeNO_50_ levels, probably due to the activation of iNOS expression through interferon-induced pathways. Moreover, virus-related FeNO_50_ elevations may be resistant to corticosteroid treatment [131] (Table 2).

### 4.3. Interpreting FeNO Levels in Asthma

The current GINA guidelines [1] recommend the use of FeNO_50_ to support, but not exclude, the diagnosis of asthma. Indeed, FeNO_50_ values can be low in some asthma phenotypes (e.g., neutrophilic asthma or asthma with obesity), while they can be elevated in non-asthma conditions (e.g., eosinophilic bronchitis, atopy, and allergic rhinitis). As a result, the diagnosis of asthma should always be performed in association with spirometry, bronchodilator reversibility/bronchial provocation tests, and clinical assessment. According to the ATS Clinical Practice guidelines [132], FeNO_50_ values <25 parts per billion (ppb) in adults (or <20 ppb in children) are considered normal, while values > 50 ppb in adults (or >35 ppb in children) are likely to indicate eosinophilic airway inflammation. Values between 25 and 50 ppb (20–35 ppb in children) should be cautiously interpreted in the clinical context (Table 3).

Nonetheless, in patients diagnosed with severe asthma, the GINA guidelines consider FeNO50 ≥ 20 ppb as a characteristic marker for identifying the type-2 inflammation phenotype, together with blood eosinophils (≥150–300 cells/µL) and sputum eosinophils (≥3%) [1]. FeNO50 has also been proposed as a useful inflammatory marker for monitoring asthma (Table 4).

In adult patients with uncontrolled moderate-to-severe asthma and a history of exacerbations in the previous year, higher blood eosinophil counts and higher FeNO_50_ levels have been associated with a greater risk of severe exacerbations [133]. However, further investigations are needed to support the predictive value of FeNO_50_ measurements in identifying a reactive and high-risk-of-exacerbation phenotype among patients with uncontrolled asthma. Considering that FeNO_50_ levels rapidly decrease when oral corticosteroids or ICSs are administered and tend to increase if ICS therapy is stopped, FeNO_50_ could instead be used as an index of poor adherence to ICS therapy, which represents a major cause of poor asthma control [134]. Moreover, FeNO_50_ values may be useful for monitoring ongoing airway inflammation and determining the responsiveness to corticosteroid treatment [135,136]. The ATS guidelines associate FeNO_50_ values <25 ppb with probable resistance to corticosteroid therapy in adults, while FeNO_50_ measurements >50 ppb are synonymous with responsiveness [136]. Accordingly, the current GINA recommendations support the decision to start ICS therapy in patients with a diagnosis or suspected diagnosis of asthma and high FeNO_50_ levels but do not suggest using low FeNO_50_ levels when deciding against treatment with ICS [1]. Low daily doses of ICSs or low doses of ICS-formoterol as a reliever are instead suggested by the GINA guidelines as the first step of the treatment to reduce the risk of severe exacerbations [1]. It has also been proposed that FeNO_50_ levels, together with other markers of disease severity and type-2 inflammation (i.e., pulmonary function tests, eosinophil count, and sputum eosinophils), play a potential role in guiding the step-up or the step-down of ICS doses. In a 2016 Cochrane meta-analysis, a therapeutic strategy of ICS dose adjustment based on FeNO50 values or sputum eosinophils in asthmatic adults determined a decrease in the frequency of exacerbations compared to a treatment based on clinical symptoms and asthma guidelines [137]. However, in a more recent 1-year multicenter clinical trial in children with asthma, the addition of FeNO_50_ levels to symptom-guided treatment did not reduce the number of severe exacerbations [138]. Finally, FeNO_50_ levels can predict the response to treatment with biological drugs in severe type-2 asthma. In particular, FeNO_50_ ≥ 25 ppb, together with an eosinophil count ≥150 cells/mcL, represents a predictive criterion of the therapeutic efficacy of omalizumab [139], dupilumab [140], and tezepelumab [140].

### 4.4. FeNO in Asthma: Challenges and Future Directions

In summary, FeNO_50_ is a non-invasive and easy-to-measure biomarker of eosinophilic bronchial inflammation. FeNO_50_ has low specificity in the diagnosis of asthma because high values can also be found in subjects affected by other inflammatory diseases, especially if they are allergic and characterized by eosinophilia (e.g., allergic rhinitis and atopic dermatitis). However, good levels of specificity and sensitivity can be achieved in the identification of asthmatic patients when FeNO_50_ is associated with other tests, such as spirometry, bronchodilator reversibility, and bronchial provocation tests. Even considering that other endogenous and exogenous factors (e.g., age, sex, smoking, and obesity) can influence FeNO_50_ values, the test can be regarded as a useful tool in the practical management of severe asthma. In addition to being a marker of type-2 inflammation functional to patient phenotyping, FeNO_50_ can predict a good response to anti-inflammatory steroid therapy and provide information on its efficacy, as well as on the degree of patient adherence. In the context of currently available biological therapies for severe asthma, high levels of FeNO_50_ are considered predictive factors of a good response to omalizumab and tezepelumab and are included in the technical data sheet among the criteria for choosing the drug dupilumab. In severe asthma, the inflammatory process seems to also involve the most distal parts of the lung; Berry et al. [141] recorded an increase in the fraction of exhaled NO of alveolar origin (at 200 mL/s) in the expired air of severe asthmatics compared to moderate asthmatics and healthy controls. However, cut-off values indicative of eosinophilic inflammation at the level of the distal airways have not yet been identified. In this regard, an important limitation is that the test for determining alveolar FeNO requires a high expiratory effort from the patient and good compliance during the execution. Consequently, this may be difficult to perform, with the impossibility of obtaining repeatable results in patients with impaired respiratory function.

If exhaled nitric oxide (FeNO) is an excellent marker of bronchial inflammation, nasal nitric oxide (nNO) could be a good marker for identifying eosinophilic nasosinusal inflammation. In particular, the nNO dosage appears particularly promising in the diagnosis and monitoring of pathologies characterized by eosinophilic nasosinusal inflammation such as allergic rhinitis and non-allergic rhinitis with eosinophilia (NARES) [142]. The nNO level can be measured by completely occluding one nostril with an inert nasal olive while the contralateral nostril is left open. The subject is then asked to breathe normally for 30 s, performing expiratory maneuvers against an oral resistance of 10 cmH_2_O. These maneuvers aim to close the palatine velum and to avoid contamination in the measurement of nasal NO by oropharyngeal and bronchial NO values. The nasal olive samples the air exhaled from the nose, and the device analyzes the amount of NO [106]. The analysis of nNO has not yet been validated, and there is a lack of reference values. Values between 450 and 900 ppb can be adopted as the normal range based on the findings of early studies conducted on healthy subjects [143]. Values below 100 ppb may be indicative of pathologies such as cystic fibrosis or primary ciliary dyskinesia [143]. In the future, it would be interesting to evaluate whether nNOS and FeNO50 levels correlate with each other in severe asthmatics with comorbid allergic rhinitis, NARES, or CRwNP and whether these values may affect the severity of respiratory symptoms presented by patients or their response to biologic therapies.

## 5. Conclusions

The incorporation of non-invasive methodologies such as the sputum cell count, breath analysis (exhaled nitric oxide fraction (FeNO)), and exhaled breath condensate (EBC) marks a significant advancement in asthma management. These non-invasive diagnostic methods offer profound insights into the mechanisms of airway inflammation, providing clinicians with valuable tools to assess disease severity, monitor treatment efficacy, and tailor therapeutic strategies. Despite challenges in standardization and the identification of universally accepted biomarkers, their potential to enhance asthma care is evident (Table 5). Future research should focus on refining protocols, validating biomarkers, and establishing comprehensive guidelines for widespread clinical adoption. By leveraging these advancements, healthcare providers can improve outcomes and quality of life for asthma patients effectively.

## Figures and Tables

**Figure 1 cells-13-01355-f001:**
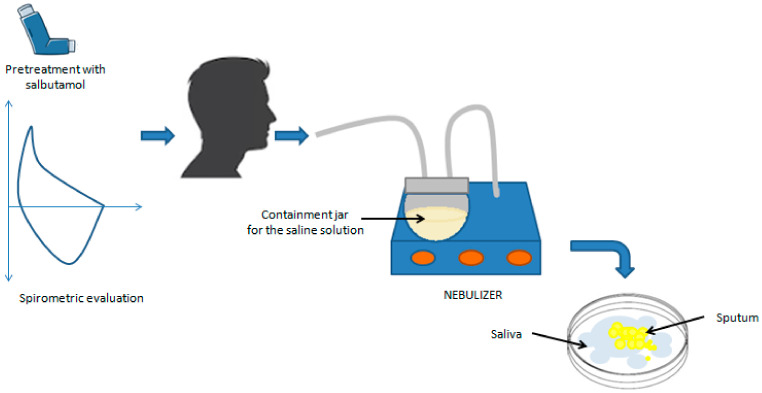
Induced sputum sampling. After spirometric evaluation and pretreatment with 200–400 μg of salbutamol, the patient inhales nebulized hypertonic saline solution, which liquefies airway secretions and allows the expectoration of respiratory secretions.

**Figure 2 cells-13-01355-f002:**
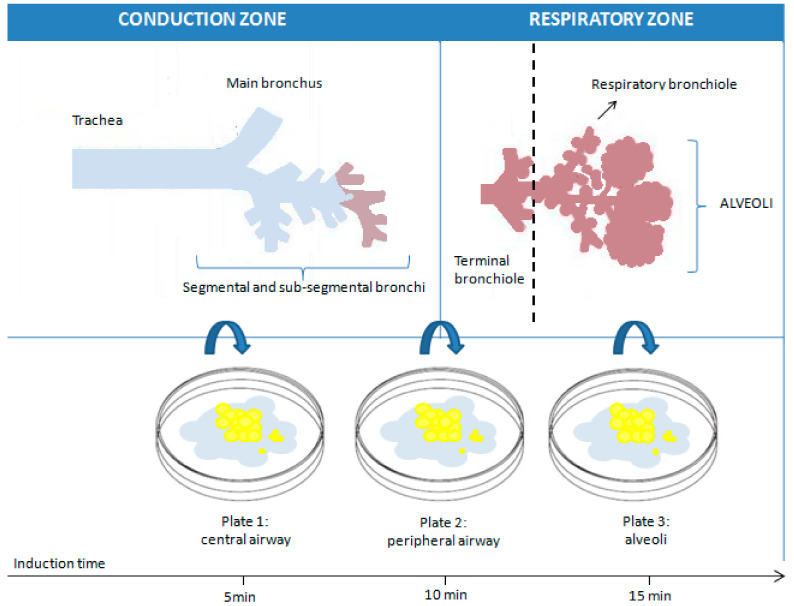
Representation of the theoretical influence of the induction time on the sampling of sputum at different levels of the respiratory tree.

**Figure 3 cells-13-01355-f003:**
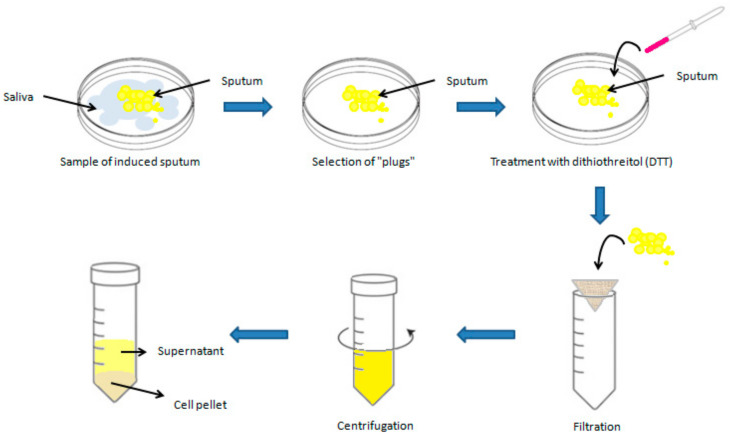
Graphical representation of induced sputum processing.

**Figure 4 cells-13-01355-f004:**
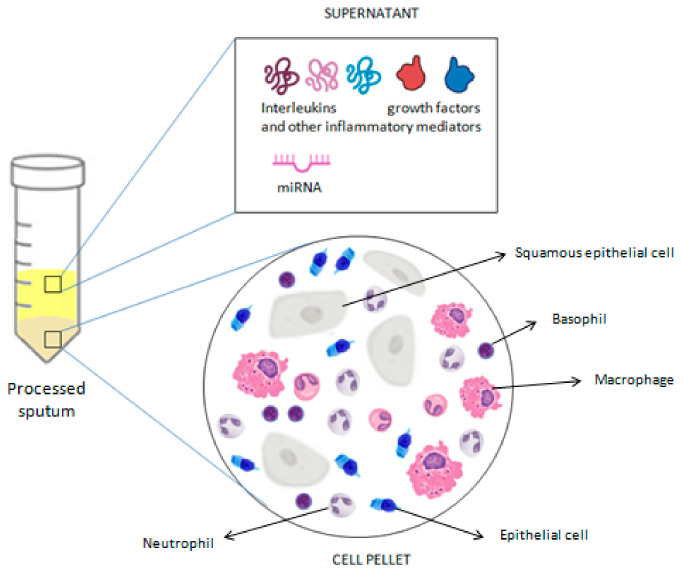
Summary of the analyses that can be performed on a processed sputum sample.

**Figure 5 cells-13-01355-f005:**
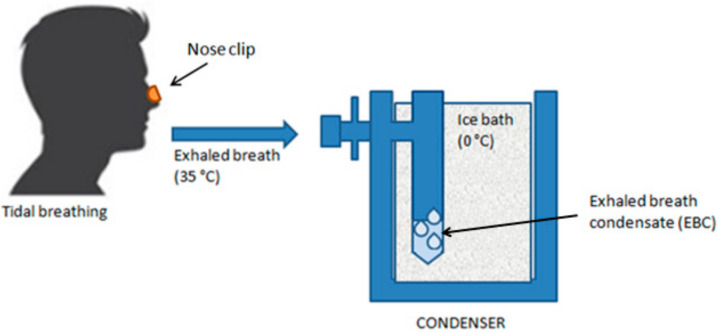
Exhaled breath condensate (EBC) sampling. The patient, wearing a soft nose clip and in a sitting position, breathes at tidal volume through the mouth into a sampling device equipped with a mouthpiece that passes into a condensing cooling chamber, in whose walls the water vapor condenses.

**Figure 6 cells-13-01355-f006:**
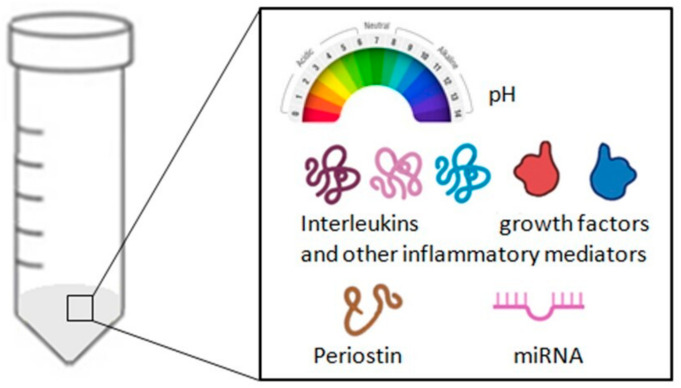
Summary of the analyses that can be carried out on a sample of exhaled breath condensate (EBC).

**Figure 7 cells-13-01355-f007:**
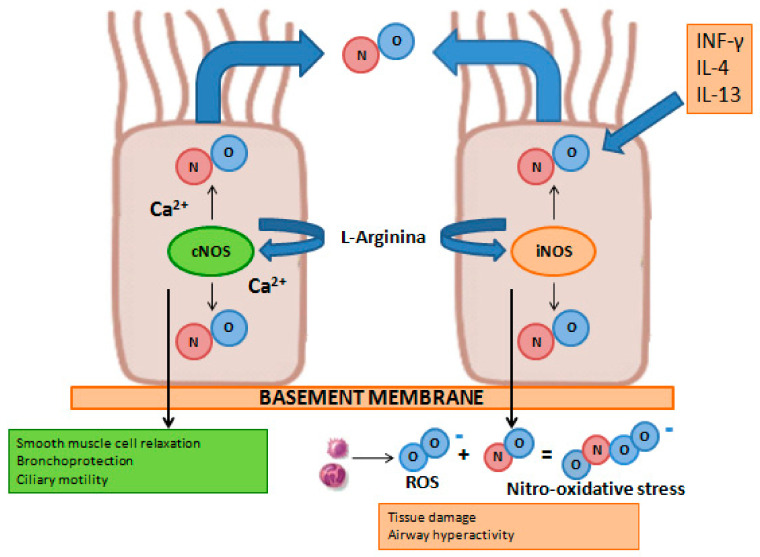
Graphical representation of iNOS and cNOS involvement on NO production under physiological (**left**) and pathological (**right**) conditions in airway epithelial cells.

**Figure 8 cells-13-01355-f008:**
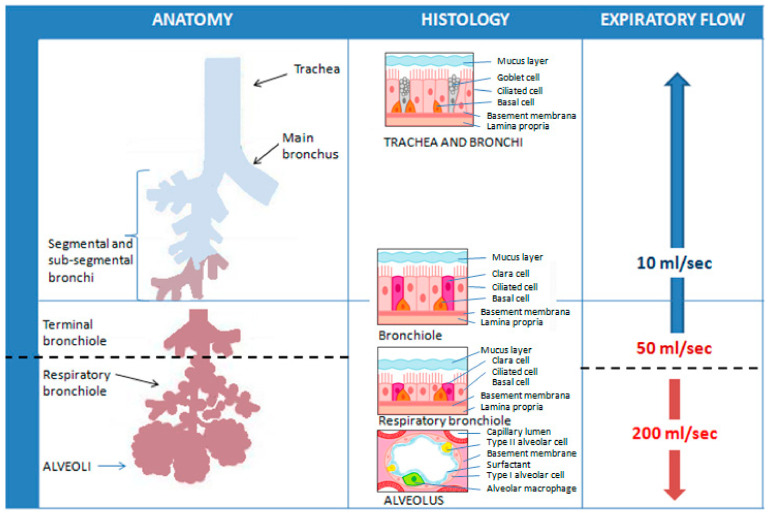
Representation of the theoretical influence of different constant expiratory flows on the sampling of FeNO values at different levels of the respiratory tree. ATS/ERS guidelines recommend an expiratory flow rate of 50 mL/s (FeNO50) as it allows targeting the lower respiratory tract as the region of interest for NO excretion.

**Table 1 cells-13-01355-t001:** Normal values and related pathological alterations for cellular composition in induced sputum.

Cell Type	Normal Range	Pathological Condition
Squamous epithelial cells (from oropharynx)	<20%	An increased percentage of squamous epithelial cells suggests salivary contamination
Epithelial cells	50–80%	An increased percentage of epithelial cells may reflect inflammation with airway damage
Macrophages	50–80%	Elevated macrophage counts may result from an ongoing immune response
Neutrophils	20–50%	Elevated neutrophil counts (>60–65%) may indicate acute inflammatory processes (e.g., bacterial infections or COPD exacerbations)
Lymphocytes	10–25%	Elevated lymphocyte counts may suggest viral infections or autoimmune disorders
Eosinophils	0–5%	Elevated eosinophil counts (>2%) may suggest eosinophilic airway inflammation

**Table 2 cells-13-01355-t002:** Endogenous and exogenous factors that potentially influence FeNO50 values.

↑ FeNO_50_	↓ FeNO_50_
Male sex	Tobacco smoke (active and passive)
Exposure to allergens and polluted air	Obesity
Menstrual cycle (proliferative or follicular phase)	Menstrual cycle (secretory or luteal phase)
Diet (rich in nitrates)	Caffeine and alcohol
Ethnicity (black and Hispanic)	Ethnicity (Caucasian)
Technical factors (environmental NO, incorrect expiratory flow, and mixed oral and nasal NO sampling)	Spirometry and physical exercise (transitory factors)
Eosinophilic/allergic asthma	Inhaled or oral corticosteroids
Allergic rhinitis	Cystic fibrosis
Chronic rhinosinusitis with nasal polyposis (CRNwNP)	Primary ciliary dyskinesia
Atopic dermatitis	Bronchiectasis
Eosinophilic bronchitis and COPD	Gastroesophageal reflux
OSAS	
Viral Infections	
Systemic lupus erythematosus	
Liver cirrhosis	

**Table 3 cells-13-01355-t003:** FeNO50 levels and assessment of the airway’s inflammation according to the ATS guidelines [116].

FeNO_50_ < 25 ppb (<20 ppb in Children)	FeNO_50_ 25–50 ppb (20–35 ppb in Children)	FeNO_50_ > 50 ppb (>35 ppb in Children)
Eosinophilic airway inflammation unlikely	Be cautious and monitor changes in FeNO_50_ over time	Eosinophilic airway inflammation present

**Table 4 cells-13-01355-t004:** Summary of the main interpretative criteria of FeNO50 values in monitoring asthma.

	Presence of Respiratory Symptoms	No Respiratory Symptoms
FeNO_50_ < 25 ppb	Evaluate alternative diagnoses	Optimal adherence to inhaled steroid Evaluate step-down
FeNO_50_ 25–50 ppb	Suboptimal adherence Inadequate therapeutic dosage Possible steroid resistance Allergen exposure	Optimal adherence and adequate therapeutic dosage Continue FeNO_50_ monitoring
FeNO_50_ > 50 ppb	Suboptimal adherence Inadequate therapeutic dosage Review inhalation technique Possible steroid resistance Allergen exposure Increased risk of exacerbation	Suboptimal adherence or inadequate therapeutic dosage Review inhalation technique

**Table 5 cells-13-01355-t005:** Summary of main advantages and disadvantages of induced sputum, EBC, and FeNO in asthma management.

	Advantages	Disadvantages
Induced sputum	Validated tool for assessment of respiratory inflammation Validated tool for identification of type-2 asthma Validated tool for monitoring of anti-inflammatory drug effectiveness Possibility to analyze multiple biomarkers	Relatively invasive Not repeatable over short time periods Contraindicated in severe obstruction Rescue medication needed Experienced personnel and specialized lab needed Time-consuming
EBC	Non-invasive Allows repeated measurements Potential tool for assessment of respiratory inflammation Possibility to analyze multiple biomarkers	Procedure awaits further validation Assays not fully reproducible Soluble markers subject to dilution Experienced personnel and specialized lab needed Expensive equipment
FeNO	Non-invasive Allows repeated measurements Validated tool for assessment of respiratory inflammation Validated tool for identification of type-2 asthma Validated tool for monitoring of anti-inflammatory drug effectiveness	Many perturbing factors Expensive equipment

## Data Availability

Data are available upon request to Soccio Piera (piera.soccio@unifg.it).

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
