# Peer review of "Breath and Sputum Analyses in Asthmatic Patients: An Overview"

_cells, 2024, doi:10.3390/cells13161355_

Round 1
Reviewer 1 Report
Comments and Suggestions for Authors
Dear Authors,
The manuscript entitled " Breath and sputum analysis in asthmatic patients: an overview" is a well prepared manucript in the field. I have only minor recomendations for thsi manuscript.
The language of the manuscript is well prepared, i do not have any comment on this.
1) I would like to ask the authors, regarding the current status of the clinical trials that are currently applied to asthmatic patients.
2) In addition, regarding the sputum analysis and the potential association of cells with disease outcome, are novel markers existed or have found, with the state of the art technology, like mass spectrometry.
3) Regarding the potentioal association of asthma and HLA, do the authors know such studies that potentially correlate the HLAs with the disease. If yes, the authors could add those studies in the main manuscript.
4) Next generation sequencing or microarray analysis could be used for the identification of microRNAs or circular DNA that could potentially be used as biomarkers for asthma.
Author Response
- I would like to ask the authors, regarding the current status of the clinical trials that are currently applied to asthmatic patients
A: Thank you for the comment. Analysis of induced sputum has only been validated for the percentage of eosinophils in the cell pellet. To this regard, we have added that the ability to identify and quantify the different cellular populations present in sputum allows physicians to assess asthma-related specific types of airway inflammation. An elevated percentage of eosinophils in induced sputum (i.e. > 2%) has been included by current GINA guidelines among criteria to identify type-2 asthma [1]. This correlation is essential for the diagnosis and monitoring of asthma, as it provides insights into the inflammatory nature of the condition and the effectiveness of anti-inflammatory drugs in reducing airway inflammation. [2, 3]. In particular, elevated levels of eosinophils in sputum have been associated with the severity of the disease and the response to treatment with inhaled corticosteroids in type-2 asthmatic patients [4]. In clinical trials evaluating the effectiveness of monoclonal agents, no reduction in sputum eosinophils was found during treatment with omalizumab [5] and mepolizumab [6] in severe type-2 asthmatics. However, Mukherjee et al. [7] have found sub-optimal responses to mepolizumab in cases of persisting sputum eosinophilia, despite a blood eosinophils count normalization. By reverse, in the real-life PROMISE study [8] a significant reduction in blood and sputum eosinophil counts was associated to an important and sustained reduction in exacerbation rate and oral corticosteroid (OCS) dose in patients with severe eosinophilic asthma receiving benralizumab. In the recent clinical trial by Svenningsen et al. [9], dupilumab reduced sputum eosinophils and CT mucus plugging, allowing a consequent improvement in MRI ventilation and asthma control. Furthermore, tezepelumab addition reduced levels of blood and sputum eosinophils and allergen-induced bronchoconstriction in a population of patients with mild allergic asthma before and after allergen challenge [10]. Overall, the use of induced sputum as a tool for evaluating airway inflammation represents a significant advancement in clinical practice, enabling more accurate diagnosis and more effective management of asthma. Clinical studies using FeNO in asthmatic patients in the monitoring of anti-inflammatory therapies have already been cited and discussed in section 4.3. The analysis of inflammatory markers in EBC has not yet been validated and there are no clinical trials on the subject.
- In addition, regarding the sputum analysis and the potential association of cells with disease outcome, are novel markers existed or have found, with the state of the art technology, like mass spectrometry.
A: Thank you for the comment. We have discussed that the combination of electrophoretic/chromatographic methods with mass spectrometry has gained growing interest for the identification of proteins whose levels change in sputum of patients with asthma. Using mass spectrometry tecnique, Lee et al. [11] found that the expression of S100 calcium binding protein A9 (S100A9) was significantly increased in the sputum of severe uncontrolled asthmatic patients compared to controlled ones and was associated to neutrophilic inflammation. According to Takahashi et al. [12] different expressed proteins in induced sputum could make it possible to distinguish between a smoking and ex-smoking severe asthma phenotype. Tariq et al. [13] found that proteins associated with gastro-oesophageal reflux disease were three- and ten-fold more prevalent in sputum of severe asthmatics compared to that of mild/moderate asthmatics and healthy controls.
- Regarding the potential association of asthma and HLA, do the authors know such studies that potentially correlate the HLAs with the disease. If yes, the authors could add those studies in the main manuscript.
A: Thank you for the comment. However, an interesting discussion about the potential association between asthma and HLA haplotypes goes beyond the scope of our review which is focused on the use of induced sputum, EBC and FeNO in the management and treatment of asthma.
- Next generation sequencing or microarray analysis could be used for the identification of microRNAs or circular DNA that could potentially be used as biomarkers for asthma.
A: Thank you for the comment. We have discussed that Several studies have revealed specific miRNA expression patterns associated with the severity and phenotype of asthma, thus providing valuable insights into the underlying biology of the disease and opening new avenues for the development of targeted and personalized therapies.
Sputum: Gomez et al. [14] identified hsa-miR-223-3p as the highest expressed microRNA in the sputum of asthmatics with increased neutrophil counts, thus suggesting it to be a key regulator of the TLR and Th17 pathways. Li et al. [15] linked an up-regulation in sputum of miR-9 expression to the mechanism of steroid resistance in asthma. According to Zhang et al. [16] a decreased expression of miR-221-3p in sputum may protect asthma patients against eosinophilic airways inflammation. Lacedonia et al. [17] found that miRNA-145, a key regulator of airway smooth muscle (ASM) function, is higher in the sputum of asthma patients compared to those of patients with different obstructive diseases. Malmhäll et al. [18] suggested a possible role of miR-155 expression in allergen-induced eosinophilic inflammation, as its expression was decreased in sputum of allergic asthmatics during pollen season. According to the suggestive hypothesis of Song et al. [19], an up-regulation of lncRNA-NEAT1 and a down-regulation of miR-128 in the sputum of children with bronchial asthma may promote airway smooth muscle remodeling.
EBC: the pattern of micro-RNAs expression in EBC of patients with asthma seems to suggest a dysregulation of the Th2 pathway with an excessive activation of Th2 cells and secretion of Th2-type cytokines including IL-4, IL-5 and IL-13 compared to patients with COPD and healthy adults [20]. The negative association found between symptomatic asthma and levels of the regulators miR-21-5p and miR-155 in the EBC of asthmatic children strengthens this hypothesis [21].
References:
- Global Initiative for Asthma (GINA). Global Strategy for Asthma Management and Prevention, 2024. https://ginasthma.org/2024-report/. Accessed June 21, 2024.
- Pizzichini E, Pizzichini MMM, Efthimiadis A, et al. Indices of airway inflammation in induced sputum: reproducibility and validity of cell and fluid-phase measurements. Am J Respir Crit Care Med. 1996;154(2):308-317. doi:10.1164/AJRCCM.154.2.8756799
- Guiot J, Demarche S, Henket M, et al. Methodology for Sputum Induction and Laboratory Processing. J Vis Exp. 2017;2017(130):56612. doi:10.3791/56612
- Jayaram L, Parameswaran K, Sears MR, Hargreave FE. Induced sputum cell counts: their usefulness in clinical practice. Eur Respir J. 2000;16(1):150-158. doi:10.1034/J.1399-3003.2000.16A27.X
- Mukherjee M, Kjarsgaard M, Radford K, et al. Omalizumab in patients with severe asthma and persistent sputum eosinophilia. Allergy Asthma Clin Immunol. 2019;15(1):21. doi:10.1186/S13223-019-0337-2
- Pavord ID, Buhl R, Kraft M, et al. Evaluation of sputum eosinophil count as a predictor of treatment response to mepolizumab. ERJ Open Res. 2022;8(2). doi:10.1183/23120541.00560-2021
- Mukherjee M, Paramo FA, Kjarsgaard M, et al. Weight-adjusted intravenous reslizumab in severe asthma with inadequate response to fixed-dose subcutaneous mepolizumab. Am J Respir Crit Care Med. 2018;197(1):38-46. doi:10.1164/RCCM.201707-1323OC/SUPPL_FILE/DISCLOSURES.PDF
- Schleich F, Moermans C, Seidel L, et al. Benralizumab in severe eosinophilic asthma in real life: confirmed effectiveness and contrasted effect on sputum eosinophilia versus exhaled nitric oxide fraction – PROMISE. ERJ Open Res. 2023;9(6). doi:10.1183/23120541.00383-2023
- Svenningsen S, Kjarsgaard M, Haider E, et al. Effects of Dupilumab on Mucus Plugging and Ventilation Defects in Patients with Moderate-to-Severe Asthma: A Randomized, Double-Blind, Placebo-Controlled Trial. Am J Respir Crit Care Med. 2023;208(9):995-997. doi:10.1164/RCCM.202306-1102LE/SUPPL_FILE/DISCLOSURES.PDF
- Gauvreau GM, O’Byrne PM, Boulet L-P, et al. Effects of an Anti-TSLP Antibody on Allergen-Induced Asthmatic Responses. N Engl J Med. 2014;370(22):2102-2110. doi:10.1056/NEJMOA1402895/SUPPL_FILE/NEJMOA1402895_DISCLOSURES.PDF
- Lee TH, Jang AS, Park JS, et al. Elevation of S100 calcium binding protein A9 in sputum of neutrophilic inflammation in severe uncontrolled asthma. Ann Allergy, Asthma Immunol. 2013;111(4):268-275.e1. doi:10.1016/j.anai.2013.06.028
- Takahashi K, Pavlidis S, Ng Kee Kwong F, et al. Sputum proteomics and airway cell transcripts of current and ex-smokers with severe asthma in U-BIOPRED: an exploratory analysis. Eur Respir J. 2018;51(5). doi:10.1183/13993003.02173-2017
- Tariq K, Schofield JPR, Nicholas BL, et al. Sputum proteomic signature of gastro-oesophageal reflux in patients with severe asthma. Respir Med. 2019;150:66-73. doi:10.1016/J.RMED.2019.02.008
- Gomez JL, Chen A, Diaz MP, et al. A Network of Sputum MicroRNAs Is Associated with Neutrophilic Airway Inflammation in Asthma. Am J Respir Crit Care Med. 2020;202(1):51. doi:10.1164/RCCM.201912-2360OC
- Li JJ, Tay HL, Maltby S, et al. MicroRNA-9 regulates steroid-resistant airway hyperresponsiveness by reducing protein phosphatase 2A activity. J Allergy Clin Immunol. 2015;136(2):462-473. doi:10.1016/j.jaci.2014.11.044
- Zhang K, Liang Y, Feng Y, et al. Decreased epithelial and sputum miR-221-3p associates with airway eosinophilic inflammation and CXCL17 expression in asthma. Am J Physiol - Lung Cell Mol Physiol. 2018;315(2):L253-L264. doi:10.1152/AJPLUNG.00567.2017/ASSET/IMAGES/LARGE/ZH50061874500009.JPEG
- Lacedonia D, Palladino GP, Foschino-Barbaro MP, Scioscia G, Elisiana G, Carpagnano. Expression profiling of miRNA-145 and miRNA-338 in serum and sputum of patients with COPD, asthma, and asthma–COPD overlap syndrome phenotype. Int J Chron Obstruct Pulmon Dis. 2017;12:1811. doi:10.2147/COPD.S130616
- Malmhäll C, Johansson K, Winkler C, Alawieh S, Ekerljung L, Rådinger M. Altered miR-155 Expression in Allergic Asthmatic Airways. Scand J Immunol. 2017;85(4):300-307. doi:10.1111/SJI.12535
- Song D, Jiang Y, Zhao Q, Li J, Zhao Y. lncRNA-NEAT1 Sponges miR-128 to Promote Inflammatory Reaction and Phenotypic Transformation of Airway Smooth Muscle Cells. Comput Math Methods Med. 2022;2022. doi:10.1155/2022/7499911
- Pinkerton M, Chinchilli V, Banta E, et al. Differential expression of microRNAs in exhaled breath condensates of patients with asthma, patients with chronic obstructive pulmonary disease, and healthy adults. J Allergy Clin Immunol. 2013;132(1):217-219.e2. doi:10.1016/j.jaci.2013.03.006
- Mendes FC, Paciência I, Ferreira AC, et al. Development and validation of exhaled breath condensate microRNAs to identify and endotype asthma in children. PLoS One. 2019;14(11):e0224983. doi:10.1371/JOURNAL.PONE.0224983
Reviewer 2 Report
Comments and Suggestions for Authors
Non-invasive diagnostic tools are very valuable for diagnosis and monitoring of conditions like asthma and allergic rhinitis. Researchers and clinicians working in the field need to be aware about their advantages and shortcomings in order to use them with maximum benefit for the diagnosis and for the patient. The manuscript provides sufficient information about the noninvasive diagnostic techniques for asthma and COPD. Perhaps the comparison between the features of the methods ascribed would be better presented in a table form which may shorten a little bit the text. I find the figures very useful.
Comments on the Quality of English LanguageI found just few misprints. In my opinion the manuscript can be accepted with minor editing corrections.
Author Response
The comparison between the features of the methods ascribed would be better presented in a table form which may shorten a little bit the text. I find the figures very useful.
A: Thank you for your suggestion. We have added a table (Table 5) summarizing main advantages and disadvantages of induced sputum, EBC and FeNO in asthma management.
I found just few misprints.
A: Thank you for the comment. English language has been improved and misprints have been corrected.
Reviewer 3 Report
Comments and Suggestions for Authors
I have read the article by Soccio et al. with great interest. The authors summarised non-invasive sampling techniques in asthma management.
Comments:
· Sputum. You need to clarify that sputum induction may induce bronchoconstriction and airway inflammation and may not be safe in people with severe airflow obstruction. Please, cite relevant articles.
· Sputum. You need to clarify that DTT may interact with commercial ELISA kits therefore the approach can limit the measurement of soluble mediators in induced sputum. Please, cite relevant articles.
· EBC. You need to clarify that the airway lining fluid is diluted by the alveolar water loss in an unpredictable ratio. You should cite relevant articles that investigated different dilution indicators, such as the measurement of electrolytes or the conductivity of vacuum evaporated or lyophilised samples.
· EBC and FENO. These are the only methods to investigate airway inflammation during the development of bronchoconstriction. A good example is exercise-induced bronchoconstriction. Changes in FENO, EBC pH, leukotrienes and adenosine were reported. Please, discuss and cite relevant articles.
· EBC. You need to clarify that condenser material and condensing temperatures may affect the content. Please, discuss and cite relevant articles.
· EBC. Please, clarify that EBC may be contaminated by the oral content. A way to assess is the measurement of amylase in EBC samples. Please, discuss the relevant articles.
Author Response
Sputum. You need to clarify that sputum induction may induce bronchoconstriction and airway inflammation and may not be safe in people with severe airflow obstruction. Please, cite relevant articles.
A: Thank you for your suggestion. We have clarified that sputum induction has the potential adverse effect to induce airway inflammation [1] and bronchoconstriction [2] in asthmatics, which can lead to even fatal asthma exacerbations in subjects with severe airflow obstruction [3].
Sputum. You need to clarify that DTT may interact with commercial ELISA kits therefore the approach can limit the measurement of soluble mediators in induced sputum. Please, cite relevant articles.
A: Thank you for the relevant comment. We clarified that analysis of sputum supernatant is generally performed using the immunoenzymatic method. However, DTT may interact with commercial ELISA kits or alter the three dimensional structure of proteins dissolved in mucus, thus limiting the measurement of soluble mediators in induced sputum. Previous studies of induced sputum from asthmatics have shown that processing with DTT may decrease detectable concentrations of. myeloperoxidase (MPO) and eosinophilic peroxidase and increase concentrations of eosinophil cationic protein (ECP), but seems to have no effect on interleukin (IL)-5 or IL-8 levels. [4, 5, 6]. These measurements are generally performed using the immunoenzymatic method. However, DTT may interact with commercial ELISA kits or alter the three dimensional structure of proteins dissolved in mucus, thus limiting the measurement of soluble mediators in induced sputum. Previous studies of induced sputum from asthmatics have shown that processing with DTT may decrease detectable concentrations of. myeloperoxidase (MPO) and eosinophilic peroxidase and increase concentrations of eosinophil cationic protein (ECP), but seems to have no effect on interleukin (IL)-5 or IL-8 levels. [4, 5, 6].
EBC. You need to clarify that the airway lining fluid is diluted by the alveolar water loss in an unpredictable ratio. You should cite relevant articles that investigated different dilution indicators, such as the measurement of electrolytes or the conductivity of vacuum evaporated or lyophilised samples.
A: Thank you for the relevant comment. We clarified that the airway lining fluid component of EBC is diluted by condensing vapor phase water in an unpredictable ratio. As result, it has assumed that to calculate the ‘‘real airway level’’ of determined mediators in EBC it is crucial to determine the so called ‘‘dilution factor’’ from each sample. To this purpose it has been suggested to use concentrations of electrolytes [7,8,9] or urea [10] or the measurement of conductance [8].
EBC and FENO. These are the only methods to investigate airway inflammation during the development of bronchoconstriction. A good example is exercise-induced bronchoconstriction. Changes in FENO, EBC pH, leukotrienes and adenosine were reported. Please, discuss and cite relevant articles.
A: Thank you for the suggestion. We discussed that the pH in EBC is reduced in cases of persistent asthma, lowered in exacerbations and correlated with disease severity and cigarette smoking [11,12,13]. An up to three log order decrease in EBC pH of asthmatic patients reflects eosinophilic inflammation and nitric oxide (NO) metabolism [14,15]. Furthermore, the development of exercise-induced bronchoconstriction was related to acute reduction of EBC pH [16]. Similarly, elevated concentrations of EBC Cys-LTs and LTB4 were found in asthmatic children during exacerbations, showing to significantly decrease after prednisone treatment [17] and EBC Cys-LT values have demonstrated to raise in asthmatic children with exercise-induced bronchoconstriction [18]. Another promising marker of airway inflammation in EBC seems to be adenosine concentration, whose value has shown to be higher in patients with worsening of asthma symptoms or with exercise-induced bronchoconstriction [19, 20]. As confirm, EBC adenosine concentration in EBC showed a positive correlation with FeNO levels in asthmatic patients [21]. Based on the above, EBC analysis and levels of exhaled fractional nitric oxide (FeNO) are the only methods allowing to in vivo investigate airway inflammation associated to the development of bronchoconstriction. However, due to the lack of standardization of the procedure, the pH, Cys-LTs levels and adenosine concentration in EBC have not been prospectively evaluated as a guide for asthma treatment, like exhaled nitric oxide and/or eosinophils in sputum. Based on these considerations, in addition to evaluating new biomarkers, it is necessary to standardize existing procedures for the introduction of EBC into clinical practice.
EBC. You need to clarify that condenser material and condensing temperatures may affect the content. Please, discuss and cite relevant articles.
A: Thank you for the relevant comment. We clarified that EBC can be collected as fluid at a condensation temperature around 0˚C or as frozen material at lower temperatures. Anyhow, it should be highlighted that the solubility of volatile mediators in the collected samples may be influenced by the collection temperature. More specifically, cold temperatures that cause the EBC to freeze may diminish the amount of volatile compounds (which are more readily absorbed into the liquid phase), while frozen storage may protect reactive or unstable compounds [22]. The collecting temperature may be clearly influenced also from that of the exhaled air
EBC. Please, clarify that EBC may be contaminated by the oral content. A way to assess is the measurement of amylase in EBC samples. Please, discuss the relevant articles.
A: Thank you for the relevant comment. We clarified that it cannot be excluded that in EBC collection there is some contamination between the saliva and the lower airways, despite studies found that the electrolyte ratios and the molecular content of EBC differ from those of saliva [7, 23]. A frequently used method to exclude oral contamination is the detection of amylase, which in EBC shows levels approximately 10,000 times lower than those in saliva [7, 19]. However, this measurement is not specific for salivary amylase and amylase can also be found in the lungs. As result, high amylase levels do not necessarily mean salivary contamination.
References:
- Gravelyn TR, Pan PM, Eschenbacher WL, Capper M, Ciffin K. Mediator Release in an Isolated Airway Segment in Subjects with Asthma. https://doi.org/101164/ajrccm/1373641. 2012;137(3):641-646. doi:10.1164/AJRCCM/137.3.641
- Smith CM, Anderson SD. Inhalation provocation tests using nonisotonic aerosols. J Allergy Clin Immunol. 1989;84(5 Pt 1):781-790. doi:10.1016/0091-6749(89)90309-6
- Saetta M, Di Stefano A, Turato G, et al. Fatal asthma attack during an inhalation challenge with ultrasonically nebulized distilled water. J Allergy Clin Immunol. 1995;95(6):1285-1287. doi:10.1016/S0091-6749(95)70088-9
- Efthimiadis A, Pizzichini MMM, Pizzichini E, Dolovich J, Hargreave FE. Induced sputum cell and fluid-phase indices of inflammation: comparison of treatment with dithiothreitol vs phosphate-buffered saline. Eur Respir J. 1997;10(6):1336-1340. doi:10.1183/09031936.97.10061336
- Louis R, Shute J, Goldring K, et al. The effect of processing on inflammatory markers in induced sputum. Eur Respir J. 1999;13(3):660-667. doi:10.1183/09031936.99.13366099
- Grebski E, Peterson C, Medici TC. Effect of physical and chemical methods of homogenization on inflammatory mediators in sputum of asthma patients. Chest. 2001;119(5):1521-1525. doi:10.1378/chest.119.5.1521
- Effros RM, Hoagland KW, Bosbous M, et al. Dilution of Respiratory Solutes in Exhaled Condensates. https://doi.org/101164/ajrccm16552101018. 2012;165(5):663-669. doi:10.1164/AJRCCM.165.5.2101018
- Effros RM, Biller J, Foss B, et al. A Simple Method for Estimating Respiratory Solute Dilution in Exhaled Breath Condensates. https://doi.org/101164/rccm200307-920OC. 2012;168(12):1500-1505. doi:10.1164/RCCM.200307-920OC
- Zacharasiewicz A, Wilson N, Lex C, et al. Repeatability of sodium and chloride in exhaled breath condensates. Pediatr Pulmonol. 2004;37(3):273-275. doi:10.1002/PPUL.10431
- Dwyer TM. Sampling airway surface liquid: Non-volatiles in the exhaled breath condensate. Lung. 2004;182(4):241-250. doi:10.1007/S00408-004-2506-3/METRICS
- Nicolaou NC, Lowe LA, Murray CS, Woodcock A, Simpson A, Custovic A. Exhaled Breath Condensate pH and Childhood Asthma. https://doi.org/101164/rccm200601-140OC. 2012;174(3):254-259. doi:10.1164/RCCM.200601-140OC
- Kostikas K, Koutsokera A, Papiris S, Gourgoulianis KI, Loukides S. Exhaled breath condensate in patients with asthma: implications for application in clinical practice. Clin Exp Allergy. 2008;38(4):557-565. doi:10.1111/J.1365-2222.2008.02940.X
- Ko FWS, Leung TF, Hui DSC. Are exhaled breath condensates useful in monitoring asthma? Curr Allergy Asthma Rep. 2007;7(1):65-71. doi:10.1007/S11882-007-0032-0/METRICS
- Hunt JF, Fang K, Malik R, et al. Endogenous Airway Acidification. Implications for asthma pathophysiology. Am J Respir Crit Care Med. 2012;161(3 I):694-699. doi:10.1164/AJRCCM.161.3.9911005
- Gaston B, Kelly R, Urban P, et al. Buffering airway acid decreases exhaled nitric oxide in asthma. J Allergy Clin Immunol. 2006;118(4):817-822. doi:10.1016/J.JACI.2006.06.040
- Bikov A, Galffy G, Tamasi L, et al. Exhaled breath condensate pH decreases during exercise-induced bronchoconstriction. Respirology. 2014;19(4):563-569. doi:10.1111/RESP.12248
- Baraldi E, Carraro S, Alinovi R, et al. Cysteinyl leukotrienes and 8-isoprostane in exhaled breath condensate of children with asthma exacerbations. Thorax. 2003;58(6):505. doi:10.1136/THORAX.58.6.505
- Carraro S, Corradi M, Zanconato S, et al. Exhaled breath condensate cysteinyl leukotrienes are increased in children with exercise-induced bronchoconstriction. J Allergy Clin Immunol. 2005;115(4):764-770. doi:10.1016/J.JACI.2004.10.043
- Huszár É, Vass G, Vizi É, et al. Adenosine in exhaled breath condensate in healthy volunteers and in patients with asthma. Eur Respir J. 2002;20(6):1393-1398. doi:10.1183/09031936.02.00005002
- Vizi É, Huszár É, Csoma Z, et al. Plasma adenosine concentration increases during exercise: A possible contributing factor in exercise-induced bronchoconstriction in asthma. J Allergy Clin Immunol. 2002;109(3):446-448. doi:10.1067/mai.2002.121955
- Huszár É, Vass G, Vizi É, et al. Adenosine in exhaled breath condensate in healthy volunteers and in patients with asthma. Eur Respir J. 2002;20(6):1393-1398. doi:10.1183/09031936.02.00005002
- Vaughan J, Ngamtrakulpanit L, Pajewski TN, et al. Exhaled breath condensate pH is a robust and reproducible assay of airway acidity. Eur Respir J. 2003;22(6):889-894. doi:10.1183/09031936.03.00038803
- Griese M, Noss J von BC. Protein pattern of exhaled breath condensate and saliva. Proteomics. 2002;2:690–696.
Round 2
Reviewer 3 Report
Comments and Suggestions for Authors
i am happy with the changes a suggest acceptance